# Key Performance Indicators for College American Football Starters: An Exploratory Study

**DOI:** 10.3390/jfmk10010019

**Published:** 2025-01-03

**Authors:** Quincy R. Johnson, Yang Yang, Dimitrije Cabarkapa, Shane Stock, Dalton Gleason, Kazuma Akehi, Dayton Sealey, Clay Frels, Douglas B. Smith, Andrew C. Fry

**Affiliations:** 1Jayhawk Athletic Performance Laboratory-Wu Tsai Human Performance Alliance, Department of Health, Sport and Exercise Sciences, University of Kansas, Lawrence, KS 66045, USA; 2Athletics Department, University of Nebraska-Kearney, Kearney, NE 68849, USA; 3Department of Kinesiology and Sport Sciences, University of Nebraska-Kearney, Kearney, NE 68849, USA; 4Department of Kinesiology, Applied Health and Recreation, Oklahoma State University, Stillwater, OK 74078, USA

**Keywords:** strength and conditioning, sport science, performance science, Wu Tsai Human Performance Alliance

## Abstract

Objectives: The purpose of this study was twofold: (a) to profile body composition and physical fitness characteristics of collegiate American football starters and (b) to examine differences in key performance indicators across position groups. These indicators included select measures of body composition, joint kinematics, as well as muscular strength and power. Methods: Sixteen National Collegiate Athletic Association (NCAA) Division-II American football athletes (age: 22.25 ± 1.1 years; height: 183.75 ± 7.8 cm; and body mass: 97.22 ± 20.39 kg) volunteered to participate in this study. A Kruskal–Wallis one-way analysis of variance by ranks test with Dunn test post-hoc adjustments was used to examine position differences between Line (*n* = 3), Big Skill (*n* = 6), and Skill (*n* = 7) position groups with α priori set at *p* < 0.05. Results: The findings of this study suggest that significant differences in body composition (*p* = 0.004), muscular strength (*p* = 0.01), and muscular power (*p* = 0.03) exist between position groups. However, no significant differences were observed in joint kinematics as assessed by the bilateral squat test (*p* > 0.05). Conclusions: Therefore, key findings from this study suggest that although significant differences in body composition, muscular strength, and muscular power exist, an emphasis should be placed on the regular assessment, development, and maintenance of optimal joint kinematics within collegiate American football populations as this appears to be a shared key performance indicator among starters.

## 1. Introduction

American football has been a popular sport in the United States for decades. The National Collegiate Athletic Association (NCAA) includes around 75,000 student-athletes who compete in football annually [1]. With advancements in technology explicitly tailored to football and the demands of the sport itself, emphasis has been placed on developing adequate evaluation and training approaches to optimize athlete performance. Body composition and physical fitness measures are commonly used within the strength and conditioning (S&C) profession to assess athletic potential, readiness to perform, and effectiveness of training, which is essential for the development of American football athletes [2,3,4].

Evaluating the football athlete is a complex task, since the sport requires a wide range of explosive football-specific movements, such as blocking and tackling, and intermittent tasks, such as sprinting and jumping [5]. Additionally, successful athletes usually possess optimal cardiorespiratory fitness, muscular endurance, upper- and lower-body muscular strength and power, the ability to sprint at maximal speed, and change direction [2,3,4,6,7]. Body composition is a standard assessment tool to assess a football player’s health status and risk for injury (especially those of the lower extremity) and is constantly monitored throughout the season [8,9,10,11,12,13]. However, further information is needed, as it relates to the NCAA Division-II American football athlete.

A recently published systematic review suggested that a higher body mass index (BMI) may be associated with a higher incidence of lower-extremity injuries [14]. However, Kaplan et al. found contradictory results [15]. With recent technological improvements, equipment such as the Dual-Energy X-ray absorptiometry (DEXA) scan and multi-frequency bioelectrical impedance analysis (BIA) can quickly assess an individual’s body composition with a high accuracy and reliability [10,16]. The InBody utilizes multiple BIA scanning athletes’ bodies for a more accurate reading than single-channel BIA systems. Additionally, they are considerably cheaper but still offer a good reliability compared to a DEXA machine, which is considered the golden standard in body composition measurements. However, it is noteworthy that BIA analyzers will constantly include systematic bias, which was noted in research focused on college athletes [16,17,18]. Nevertheless, BIA systems are still considered popular for field use among S&C staff.

Joint kinematics is a fundamental component of dynamic sporting actions. For instance, the Functional Movement Screen (FMS) and Y balance test (YBT) are popular balance and stability tests and have both been used by S&C staff to assist in injury prevention and assess training effectiveness [13,19,20,21,22]. However, new technology, such as markerless motion capture systems, can assess flexibility, balance, and movement quality within one test [23,24,25,26]. The isometric mid-thigh pull (IMTP) test can assess muscular strength, while muscular power can be assessed through the utilization of jumping tasks, most commonly, the countermovement jump (CMJ) [27,28,29,30,31]. The IMTP has become popular because it often requires less time and provides more information on specific muscular strength characteristics. IMTP performance has also been found to be positively correlated with traditional tests of strength, sprint speed, and agility [32,33,34,35]. Both muscular strength and power can also be tested on force plates, notably, lower-body strength and power, with early work in American football occurring in 2002 at The College of New Jersey and replicated in various research across different sports to monitor athletes’ neuromuscular readiness [27,28,29,30,31,36,37,38,39,40]. Prior research suggests that joint kinematics, muscular strength, and muscular power significantly differ based on health status as well as playing stats and the role on the team [21].

Body composition and physical fitness measures are commonly used within the S&C profession to assess athletic potential, readiness to perform, and the effectiveness of training. However, limited information is available on NCAA Division II American football athletes. Therefore, the primary and secondary purpose of this study was to profile the body composition and physical fitness characteristics of NCAA Division II American football athletes, and to examine differences in general measures of body composition, joint kinematics, muscular strength, and muscular power across position groups to identify key performance indicators (KPIs). Based on the demands of the sport and general attributes of athletes, the authors hypothesize that significant differences will be observed between position groups in the measures of body composition, joint kinematics, muscular strength, and muscular power.

## 2. Materials and Methods

### 2.1. Study Design

This study used a cross-sectional design to examine the body composition, joint kinematics, muscular strength, and lower-body muscular power characteristics of NCAA Division-II American football athletes. Prior to the beginning of the 2022 season, the subjects participated in a voluntary performance testing battery. According to McKay et al., this cohort of athletes would be classified as “Tier 3: Highly Trained/National”, which only includes approximately 0.014% of the global population [41].

### 2.2. Participants

Sixteen NCAA Division-II American football starters (age: 22.25 ± 1.1 years; height: 183.75 ± 7.8 cm; and body mass: 97.22 ± 20.39 kg) participated in the study. All the subjects were free of musculoskeletal injuries, and prior to data collection, they regularly participated in training sessions administered by their respective S&C coaches. Training sessions occurred three days per week and included exercises focused on developing football-specific muscular strength and power, explosiveness, sprinting speed, and agility. The athletes typically performed barbell squat, press, and deadlift variations to develop muscular strength. Meanwhile, they also performed Olympic weightlifting variations, plyometrics, and ballistic exercises to develop muscular power and explosiveness. The subjects who did not meet the criteria of being free of musculoskeletal injuries, regularly participating in training sessions, and being a starter were excluded from this study. The testing procedures performed in this study were approved by the University Institutional Review Board and the participants provided consent (031022-1).

### 2.3. Testing Battery

The performance testing battery was developed through collaboration between coaching, S&C, sports medicine, and academic professionals to provide a profile of the body composition and physical fitness characteristics of offensive and defensive starters to support athlete health, well-being, and performance, as well as to better understand potential KPIs. Data were collected 4–5 days prior to the start of practice for the 2022 football season by trained professionals. The subjects were separated into offensive and defensive groups. Athletes within the offensive group were assessed on the first day (0815 h while the defensive group was assessed on the second day (0900 h). For each group, testing was completed within one day in the following order: body composition, joint kinematics, muscular strength, and power assessments. The analysis procedures encompassed the data only from the athletes that completed all the relevant tests. Upon arrival at the athletic facility, the subjects were familiarized with the testing procedures before having their body composition assessed.

### 2.4. Body Composition Testing

First, height was measured with a standard stadiometer (Cardinal; Detecto Scale Co., Webb City, MO, USA), and then, body mass characteristics were measured utilizing a bioelectrical impedance analyzer (InBody 270, Cerritos, CA, USA). Measurements were rounded to the nearest 0.01 cm and 0.01 kg. Prior to testing and in agreement with InBody’s recommendations, it was recommended that the subjects avoid strenuous exercise (6–12 h prior), eating (3–4 h), consuming alcohol (24 h), using a shower or sauna, and using lotion or ointment on their hands or feet. Additionally, the participants were encouraged to hydrate well (12–24 h), stand upright for at least 5 min, use the bathroom, and remove socks, heavy clothing, and metal objects prior to testing in alignment with prior research [16,17,18]. The body composition testing required approximately three minutes of time per person.

### 2.5. Joint Kinematics, Flexibility, and Stability Testing

Joint kinematics were assessed via a markerless three-dimensional movement screen (DARI Motion, Overland Park, KS, USA) that for the aim of this study assessed the functionality of the lower extremities during a bilateral squat movement [23,24,25,26,42,43]. The subjects were instructed to stand at the center of the movement screening platform with their feet in a comfortable position (inside or at shoulder width with slightly externally rotated or straight foot position) to perform the squat movement, while maintaining an upright trunk position and with both arms and hands placed in front of the body. On the verbal command, the subjects were instructed to perform the body squat as deep as they could and then return to the starting position. The subjects performed two body squat trials, and the second trials were captured and analyzed. The joint kinematics testing required approximately eight minutes per person including a five-minute stationary cycle warmup session.

### 2.6. Warmup and Familiarization

The subjects then performed a standardized warmup protocol administered by a National Strength and Conditioning Association-approved Certified Strength and Conditioning Specialist prior to the muscular strength and power testing. All the phases of the warmup served specific purposes to adequately prepare the athletes for testing. The warmup lasted 15 to 20 min and consisted of three parts: movement preparation, a dynamic warmup, and a potentiation segment. The movement preparation segment consisted of five mobility and activation exercises. The ankle, hip, shoulder, and spine were addressed during this phase. During the dynamic warmup, the athletes were instructed to perform 10 repetitions of 10 exercises with a PVC pipe. The exercises included body weight squats, lunges, presses, and hip hinges. The warmup protocol was designed to move throughout all three planes of motion and raise body temperature. The potentiation segment of the warmup consisted of three different types of jumps. The first jump was a series of hops ascending in height to increase stiffness. The second jumping exercise was a series of three consecutive CMJs designed to increase the rate of force development. The final jump exercise was two reps of a standard CMJ at submaximal intensity to prepare for the series of tests performed. Ground reaction forces for each type of assessment were sampled at 1000 Hz using a wireless uniaxial dual force plate system (Hawkin Dynamics Inc., Westbrook, ME, USA).

### 2.7. Isometric Strength Testing

Muscular strength was assessed via the IMTP. The subjects were instructed to stand in the center of the force plates with their feet slightly inside shoulder width, an upright chest, a neutral spine with their head facing forward, and their hands aligned in a comfortable position with the barbell height set at or around the mid-thigh area [33,34,35,44,45,46]. A standard 45lb. barbell (York Barbell, York, PA, USA) was positioned underneath safety racks secured to a weight rack (Samson Equipment, Las Cruces, NM, USA). On the researcher’s verbal command (e.g., 3-2-1 pull, pull, pull), the subjects were instructed to pull the barbell against the safety rack as quickly and forcefully as their ability would allow. The duration of the IMTP was approximately 2–3 s. A ramp-up protocol was implemented to prepare the subjects for maximal effort repetitions. First, one practice repetition of the IMTP was completed at a self-perceived 50% maximal effort, then another practice repetition of the IMTP was completed at a self-perceived 75% maximal effort. Then, two maximal effort repetitions of the IMTP were completed and utilized for the final data analysis. The participants were allowed to rest between efforts until they felt recovered. An estimation of the rest period utilized within this protocol would be 15–20 s between repetitions. The muscular strength testing required approximately four minutes of time per person.

### 2.8. Countermovement and Multi-Rebound Jump Testing

Muscular power was assessed via the CMJ with an arm swing as well as a multi-rebound (MR) test without an arm swing. For the CMJ assessment, the subjects were instructed to stand in the center of the force plates with a self-selected foot position and their arms above their head. On the researcher’s verbal command (e.g., 3-2-1 jump), subjects were instructed to jump as high as they could as fast as they could. The depth of the squat, knee flexion, and amount of arm extension used during the CMJ was determined by each participant. For the MR assessment, the subjects began by standing in the center of the force plates with a self-selected foot position with their hands on their hips and thumbs tucked into their waistband. On the researcher’s verbal command (e.g., 3-2-1 go), the subjects were instructed to jump as high as they could as fast as they could five times without pause. The depth of the squat and knee flexion used during the MR was determined by each participant [42]. A ramp-up protocol was utilized to prepare the subjects for maximal effort repetitions. First, one practice repetition of the CMJ was completed at a self-perceived 50% maximal effort, followed by one practice repetition of the CMJ completed at a self-perceived 75% maximal effort. Then, two maximal effort repetitions of the CMJ were completed and utilized for the final data analysis. An identical approach was utilized for the MR test. The participants were allowed to rest approximately 15–20 s between efforts until they felt recovered. This rest period agrees with previous research [24]. The muscular power testing required approximately four minutes of time per person.

### 2.9. Statistical Analyses

Descriptive statistics, means, and standard deviations were calculated for each variable. Shapiro–Wilk’s test corroborated that the assumption of normality was not violated for any of the dependent variables examined in the present study. Based on the sample size and composition, a Kruskal–Wallis one-way analysis of variance by ranks test with Dunn test post-hoc adjustments was used to examine the position group-specific differences in body composition, joint kinematics, muscular strength, and muscular power metrics among Line (n = 3), Big Skill (n = 6), and Skill (n = 7). Hedges’ g was used to calculate the measure of the effect size [i.e., g = 0.2 is a small effect, g = 0.5 is a moderate effect, and g > 0.8 is a large effect] [47]. The statistical significance was set α priori to *p* < 0.05. A priori power analysis using G*Power software 3.1.9.7 version was conducted to determine the necessary sample size. Based on a medium effect size (Hedges’ g = 0.5) and a desired power of 0.80 at a significance level of 0.05, the calculated sample size was 159 participants, which is uncommon for this demographic. However, when the post hoc achieved power was calculated for each measure of significance based on the sample size, findings from the current sample achieved the desired power of 0.80 at a significance level of 0.05. To account for the sample size and to ensure accuracy, the reported significance values were adjusted by the Bonferroni correction for multiple tests. All statistical analyses were completed with SPSS (Version 26.0; IBM Corp., Armonk, NY, USA). KPI data can be referenced in Appendix A.

## 3. Results

Descriptive statistics for each dependent variable are presented in Table 1 and Table 2. As hypothesized, significant differences were observed among the position groups for specific measures of body composition, muscular strength, and muscular power. The Linemen and Big Skill position groups weighed significantly more than the Skill groups, and the Big Skill group produced more muscular force within the 250 ms time epoch when compared to skill. However, the Skill position group jumped higher, faster, and had a shorter braking phase when compared to Linemen, but these values were similar to Big Skill groups. No significant differences were observed for any measure of joint kinematics (*p* > 0.05) reported in this study. Although non-significant, these findings may be meaningful for understanding the KPIs of American football starters beyond body composition, muscular strength, muscular power, sprinting speed, and agility.

## 4. Discussion

This study was conducted to identify KPIs within a cohort of NCAA Division-II American football starters, and the results provide further information on position group-specific differences. Unique differences in body composition, muscular strength, and muscular power were observed among the position groups. However, no significant differences in lower-extremity joint kinematics as assessed by the bilateral squat test were observed. Previous research on NCAA Division-I and II American football athletes has reported similar findings for similar measures of body compositions and physical fitness [3,16,18,33,34,39]. Combined with these studies, the current findings provide detailed insights into the KPIs of athletic performance for the collegiate American football athlete, irrespective of the competition level.

It could be expected that body composition measures would differ among position groups based on the demands of the sport, characteristics that are unique to playing positions (i.e., athlete needs), and findings reported in prior investigations [2,3,4,12,18,48]. For instance, Melvin et al. observed significant differences in the muscle cross-sectional area of the vastus lateralis, LBM, PBF, and FM among NCAA Division-I linemen (LBM; 96.4 kg, PBF; 23.4%, FM; 30.9 kg) and skill position groups (LBM; 74.5 kg, PBF; 15.3%, FM; 14.3 kg) [48]. In this study, all the measures of body composition were significantly greater in Line when compared to Skill position groups (Table 1), which agree with findings mentioned previously. However, specific measures of body composition such as DLM, LBM, SMM, BMI, and PBF did not significantly differ between the Big Skill and any other position group included within this study. These findings are likely due to position-specific body composition requirements within the sport of American football and may also be useful for creating tactical advantages. Furthermore, these findings may be the first to provide objective data regarding an optimal body composition range for NCAA Division-II American football athletes who are starters. However, further research is required to examine specific relationships among position groups, body composition, and muscular strength and power qualities to better understand similarities and differences among position groups.

Lower-extremity joint kinematics as assessed by the bilateral squat test did not differ among the position groups in this sample. Although research has reported performance on similar tests of joint kinematics utilizing the FMS and Y-Balance tests, this study is the first to examine such with the utilization of DARI motion within the NCAA Division-II American football population [39,49,50]. These findings are unique and highlight the importance of foundational movement mastery regardless of position group, and the need for a well-designed S&C program that emphasizes the quality of exercises being performed. Furthermore, should an athlete experience an injury, reestablishing general as well as sport-specific joint kinematics during the return to play and return to performance phases should be of keen interest for the sports performance staff based on the similarity of this KPI across position groups and within this cohort of starters. Athletes who participated in this study trained in a strength and conditioning system which focused on physical development through Olympic (i.e., clean, snatch, and jerk) and ground-based resistance training exercises (i.e., squat, bench press, and deadlift). Additionally, velocity-based training and an emphasis on training all contraction types (i.e., eccentric, isometric, and concentric) were key components of this system as well. Altogether, each component, when utilized with the correct volumes and intensities, may have contributed to the homogeneity of lower-extremity joint kinematics observed within this study.

Within the current cohort, specific measures of muscular strength and power differed among the position groups. Interestingly, the measures of peak force and relative peak force did not significantly differ among the position groups. This further emphasizes the importance of developing absolute strength and explosive strength regardless of the position group within the American football population and agrees with prior findings in NCAA Division-I, II, and III populations [3]. However, further research is needed to examine specific contributors to differences in muscular power as assessed by CMJ and MR performance within the American football population. Although Skill position groups jumped significantly higher and more efficiently transferred GRFs than Line in both conditions, no significant differences in jump strategy (e.g., countermovement depth, unweighting, propulsive phase, etc.) were observed among any position group, except for the braking phase which may be considered a KPI for starters. Beyond differences in body mass and technical ability, differences and similarities in jump height between position groups may be attributed to the significantly shorter braking phase and significantly greater takeoff velocity of the Skill position group during the CMJ when compared to the Line position group. For sport performance professionals, this information may be valuable and can be utilized in the physical development of specific physical characteristics that contribute to athletic performance for college football starters.

To the best of our knowledge, this is one of the few available studies examining body composition and physical fitness characteristics to this degree at the NCAA Division-II level. However, this study is not without limitations. The lack of the ability to record the athlete’s previous injury history and training status, as well as the sample size, should be acknowledged as limiting factors. In addition, further research is warranted to examine relationships between dependent variables and whether the findings of this study apply to other levels of competitive football.

## 5. Conclusions

The results from this investigation provide a profile of KPIs specific to NCAA Division-II American football starters. Furthermore, the methods and findings reported can provide practitioners with actionable information specific to the sport, level of competition, and position group that can be utilized to identify KPIs relevant to their team. Once identified, the information gathered from the testing battery can be applied to the development of a comprehensive training program targeted at addressing and improving these qualities that drive distinct outputs for each performance variable with the hope of enhancing sport performance. In this case, physical characteristics such as body composition, kinematics, muscular strength and power, and reactivity should be prioritized when attempting to improve training approaches and maximize athletic performance.

The authors also hope that these results help S&C professionals, sport coaches, sport scientists, sports medicine professionals, nutritionists and registered dietitians, and other members of the sport performance team to understand the physical profiles of collegiate American football athletes to further drive their decisions on and off the field of play. A high-performance model such as this will be dependent on the resources available and will vary based on the level of competition. At the lower levels, fluid communication amongst departments is crucial to sharing institutional resources so that detailed information, like the findings in this study, can be collected and applied appropriately for the athlete’s benefit. It is the author’s hope that the information from the present study lays the groundwork for more high-performance opportunities at all levels of competition with the goal of promoting interdisciplinary collaboration to develop systematic, evidence-based approaches for assessing KPIs to better support athlete health, wellness, and performance.

## Figures and Tables

**Table 1 jfmk-10-00019-t001:** Descriptive statistics, means, and standard deviations ($\bar{x}$ ± SD) for body composition and joint kinematics.

Variable	Line	Big Skill	Skill	*p*	*g*
Demographics and Anthropometrics					
Age (years)	22.33 ± 0.58	22.67 ± 1.03	21.86 ± 1.35	0.445	0.31
Height (cm)	192.19 ± 3.88 *	183.73 ± 6.76	180.16 ± 7.74	0.049	1.08
Weight (kg)	131.60 ± 12.13 *	99.93 ± 3.91 *	80.16 ± 6.35	0.002	4.32
Body mass index (kg/m^2^)	35.73 ± 4.45 *	29.68 ± 1.94 *	24.74 ± 1.90	0.002	2.61
Body Composition					
Dry lean mass (kg)	25.97 ± 0.65 *	22.78 ± 1.63	19.22 ± 1.51	0.004	2.87
Body fat mass (kg)	35.56 ± 13.35 *	15.60 ± 4.58	8.75 ± 3.30	0.004	2.38
Lean body mass (kg)	96.01 ± 2.25 *	84.33 ± 5.55	71.42 ± 5.53	0.004	2.94
Skeletal muscle mass (kg)	55.60 ± 1.14 *	48.72 ± 3.37	41.24 ± 3.43	0.004	2.81
Percent body fat (%)	26.53 ± 7.63 *	15.60 ± 4.67	10.87 ± 3.78	0.015	1.86
Kinematics					
Squat depth (cm)	55.71± 9.38	59.65 ± 6.30	63.87 ± 7.19	0.238	−0.67
Weight distribution left (%)	51.05 ± 1.22	50.24 ± 1.36	49.88 ± 1.79	0.388	0.46
Weight distribution right (%)	48.95 ± 1.22	49.76 ± 1.36	50.14 ± 1.77	0.348	−0.47

Note: * = significantly different when compared to skill (*p* < 0.05); g = 0.2 is a small effect, g = 0.5 is a moderate effect, and g > 0.8 is a large effect.

**Table 2 jfmk-10-00019-t002:** Descriptive statistics, means, and standard deviations ($\bar{x}$ ± SD), for muscular strength, power, and reactivity.

Variable	Line	Big Skill	Skill	*p*	*g*
Muscular Strength					
Peak force (N)	3993.17 ± 703.31	3745.00 ± 273.74	3172.95 ± 562.35	0.071	0.97
Peak relative force (%)	250.33 ± 32.17	276.42 ± 24.42	296.47 ± 56.01	0.266	−0.70
Force at 250 ms (N)	2978.33 ± 368.48	2941.31 ± 357.95 *	1993.67 ± 464.48	0.011	1.40
RFD 0–250 ms (N/s)	5395.33 ± 1323.87	6158.89 ± 1362.18 *	3520.48 ± 1761.21	0.033	0.69
Muscular Power					
Jump height (cm)	37.78 ± 6.18 *	55.20 ± 5.48	58.14 ± 4.46	0.023	−2.33
Countermovement depth (cm)	−49.13 ± 23.65	−41.20 ± 8.76	−43.24 ± 11.52	0.805	−0.22
Unweighting phase (%)	45.00 ± 8.80	47.88 ± 6.53	54.97 ± 4.58	0.053	−1.02
Braking phase (%)	24.37 ± 1.08 *	20.66 ± 3.37	17.79 ± 3.02	0.034	1.48
Propulsive phase (%)	30.64 ± 7.77	31.46 ± 3.42	27.24 ± 2.89	0.146	0.61
Takeoff velocity (m/s)	2.72 ± 0.22 *	3.29 ± 0.16	3.37 ± 0.13	0.023	−2.36
Time to takeoff (s)	1.12 ± 0.15	0.96 ± 0.12	1.02 ± 0.17	0.404	0.42
RSI-modified (ratio)	0.34 ± 0.08 †	0.59 ± 0.09	0.60 ± 0.15	0.031	−1.81
Reactivity					
Mean jump height (cm)	23.77 ± 5.35 *	29.90 ± 7.22	36.22 ± 5.57	0.020	−1.26
Mean RSI (ratio)	2.10 ± 0.20 *	2.24 ± 0.34	2.79 ± 0.31	0.017	−1.4
Mean force (N)	3859.97 ± 218.11 *	3149.94 ± 168.90	2959.59 ± 322.79	0.018	2.27
Mean of top three jumps (cm)	25.07 ± 4.92 *	31.38 ± 7.36	37.58 ± 5.53	0.024	−1.28
Mean RSI of top three jumps (cm)	2.16 ± 0.28	2.30 ± 0.34	2.82 ± 0.27	0.022	−2.55

Note: RSI = reactive strength index; † = significantly different when compared to big skill (*p* < 0.05); * = significantly different when compared to skill (*p* < 0.05); and g = 0.2 is a small effect, g = 0.5 is a moderate effect, and g > 0.8 is a large effect.

## Data Availability

The data presented in this study are available upon request from the corresponding author.

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
