# Peer review of "Key Performance Indicators for College American Football Starters: An Exploratory Study"

_jfmk, 2025, doi:10.3390/jfmk10010019_

Round 1
Reviewer 1 Report
Comments and Suggestions for Authors
My recommendations are as follows:
Abstract – I recommend mentioning the average age and standard deviation of the subjects. I recommend mentioning the number of subjects in the two groups targeting the positions specific to the American football game.
Participants – I recommend calculating the sample power by calculating G Power.
Line 207 mention tables 1-5, and the presentation of the results only mentions tables 1 and 2 I recommend clarifications.
Lines 216-217 I recommend that you bold the text.
Results – I recommend that you expand the interpretation of the results.
I recommend that under tables 1 and 2 you mention descriptively what the statistical indicator g represents.
Lines 226-227 I recommend clarifications regarding the mentions in parentheses.
At the end of the Discussion section I recommend mentioning the practical implications of the study. I also recommend expanding the limitations.
I recommend rewriting lines 281-293, they are too general and not all the aspects mentioned in the study are evident. I recommend rewriting the conclusions focused on the results.
Reviewer 2 Report
Comments and Suggestions for Authors
Introduction
It is recommended that the information provided on the variables of physical condition in relation to football and its importance for performance be expanded. While the analysis is a valuable contribution, the importance of physical condition in football players is somewhat lacking.
Lines 54-58 discuss this topic, yet do not cite the sources that consider it a significant aspect. To enhance the credibility of the text on InBody, it is essential to either cite the sources or provide a detailed explanation of why this information is important.
Materials and Methods
En principio falta explicar los entrenamientos previos que han realizado los jugadores antes del estudio y si su entrenamiento ha sido el mismo o diferenciado por puesto, ya que esto puede afectar a los resultados del estudio y es un indicador importante para el desarrollo y diseño de su estudio.
It is essential to provide a detailed account of the players' previous training regimen, including whether their training was uniform or differentiated by position. This information is crucial for interpreting the study's outcomes and for designing future studies.
It is essential to delineate the type of sampling employed in the study and to cite the relevant literature on the subject.
The testing battery should be described in detail, including the background, experience and characteristics of the subjects who have taken the data.
It is essential to elucidate the manner in which the participants became acquainted with the testing procedures. Was this familiarity limited to the assessment of body composition, or were they permitted to view, comprehend, or attempt the remaining tests in advance?
Explain whether the body composition testing was based on scientific evidence or a protocol, or if it was a self-developed design. If the latter is the case, provide an explanation of the reliability of the tests.
Warm-up and familiarisation: provide details of the specialist's training, experience and qualifications.
Countermovement and multi-rebound jump testing: clarify how the percentage was measured, as this affects the subsequent data. Without this information, the data is open to subjectivity and therefore the methodology should be modified.
Results: It seems plausible to suggest that this is a typographical error, given that the document contains references to tables 1-5, whereas only tables 1-2 are present. Furthermore, the authors do not elucidate the discrepancies by categorising the players according to their positions. In the event of tables being absent, they should be incorporated into the main body of the paper, as they are essential for understanding the discrepancies between playing positions, which are then included in the discussion and results. However, this is not indicated.
Discussion
This section should be expanded with additional relevant scientific evidence. While not from the same category of competition, studies on this sport analyse some of the variables studied and will help to gain a deeper understanding of the objective of their study and the differences with other levels of competition. This would considerably enhance the study's value and relevance within the scientific community.
Lines 236-238 and 260-265 should be incorporated into the discussion section, rather than the main body of the text, as they pertain to future research or the limitations of the study.
Conclusions:
Due to the limitations that you yourself point out, the conclusion cannot be so certain, but should be nuanced and not be taken as a certainty that all teams can have the same results when there are several limitations that can change the results. The conclusion should be rephrased in a way that is more appropriate to your study and the sample used.
Reviewer 3 Report
Comments and Suggestions for Authors
General comments
The purpose of this study was twofold: a) to profile body composition and physical fitness characteristics of collegiate American football starters; b) to examine differences in key performance indicators across position groups. This is a very interesting study, but there are some critical points from a conceptual and methodological point of view that need to be corrected.
Specific comments:
Title: Key Performance Indicators usually refers to technical and tactical (behavioral) variables. I would also remove starters, because there is no comparison between starters and non-starters.
Key words: Some key words should be reworded. “Wu Tsai Human Performance Alliance“, ‘football strength and conditioning’ (only football because it already repeats S&C); football sport Science (”repeats football).
Introduction: this section is described in narrative form, but you should add some evidence and data on body composition, joint kinematics, muscular strength, and muscular power.
Materials and Methods:
- Please clarify the power of the sample and the inclusion and exclusion criteria. Also, add the ethical approval.
- Testing Battery: clarify the application procedures (timing and methodological procedures).
- Body composition testing: indicate the rounding values for the measurement you used (0.1 cm ?). In addition, the reference values must be reported.Also add the cut-off values.
Results: this section is presented clearly and concisely. I only recommend presenting some figures to make the interpretation of the results more attractive.
Discussion:
- The results are not discussed in depth.You should present the interpretation of the results point by point.
- The practical applications, future prospects and limitations of the study should be presented.
Conclusion:
- This section should be short and concise and should report only the main outcome of the study and practical application and/or limitation of the study.
